# Ancient DNA Reveals Maternal Philopatry of the Northeast Eurasian Brown Bear (*Ursus arctos*) Population during the Holocene

**DOI:** 10.3390/genes13111961

**Published:** 2022-10-27

**Authors:** Eugenia Boulygina, Fedor Sharko, Maksim Cheprasov, Maria Gladysheva-Azgari, Natalia Slobodova, Svetlana Tsygankova, Sergey Rastorguev, Lena Grigorieva, Martina Kopp, Jorge M. O. Fernandes, Gavril Novgorodov, Gennady Boeskorov, Albert Protopopov, Woo-Suk Hwang, Alexei Tikhonov, Artem Nedoluzhko

**Affiliations:** 1Kurchatov Center for Genomic Research, National Research Centre “Kurchatov Institute”, 123182 Moscow, Russia; 2Limited Liability Company ELGENE, 109029 Moscow, Russia; 3Research Center of Biotechnology of the Russian Academy of Sciences, 119071 Moscow, Russia; 4Laboratory of P.A. Lazarev Mammoth Museum of the Research Institute of Applied Ecology of the North, North-Eastern Federal University Named after M. K. Ammosov, 677000 Yakutsk, Russia; 5Genomics Division, Faculty of Biosciences and Aquaculture, Nord University, 8049 Bodø, Norway; 6Institute of Diamond and Precious Metals Geology, Siberian Branch of Russian 5 Academy of Sciences, 677007 Yakutsk, Russia; 7Academy of Sciences of Sakha (Yakutia), 677007 Yakutsk, Russia; 8UAE Biotech Research Center, Abu Dhabi 30310, United Arab Emirates; 9Zoological Institute Russian Academy of Sciences, 190121 Saint-Petersburg, Russia; 10Paleogenomics Laboratory, European University at Saint Petersburg, 191187 Saint-Petersburg, Russia

**Keywords:** brown bear, *Ursus arctos*, Holocene, ancient DNA, mitogenome, sequencing, Novosibirsk Islands, Yakutia, philopatry, mitochondrial inheritance

## Abstract

Significant palaeoecological and paleoclimatic changes that took place during Late Pleistocene—Early Holocene transition are considered important factors that led to megafauna extinctions. Unlike many other species, the brown bear (*Ursus arctos*) has survived this geological time. Despite the fact that several mitochondrial DNA clades of brown bears became extinct at the end of the Pleistocene, this species is still widely distributed in Northeast Eurasia. Here, using the ancient DNA analysis of a brown bear individual that inhabited Northeast Asia in the Middle Holocene (3460 ± 40 years BP) and comparative phylogenetic analysis, we show a significant mitochondrial DNA similarity of the studied specimen with modern brown bears inhabiting Yakutia and Chukotka. In this study, we clearly demonstrate the maternal philopatry of the Northeastern Eurasian *U. arctos* population during the several thousand years of the Holocene.

## 1. Introduction

The brown bear (*U. arctos*) is a widespread species inhabiting the forest zone of Eurasia, including Northeast Asia (Yakutia, Chukotka, and Kamchatka Peninsula) [1]. The association with forest habitats explains why Pleistocene findings of *U. arctos* are rare in the northern part of Eastern Siberia, where open tundra–steppe and steppe landscapes prevailed during the Pleistocene. Over 60% of the modern distribution range of brown bears falls within Russia; however palaeoecological and palaeobiological data from the region remain rare [1,2,3]. Genetic data, including complete modern Eurasian brown bear mitogenomes, are abundant in the literature, but Pleistocene brown bear mitochondrial genomes are scarce [3,4,5,6,7].

Brown bears have inhabited Northeast Eurasia since at least the Middle Pleistocene, 300–400 thousand years ago (kya) [8]. Pleistocene fossils of brown bears have been found across the Yakutia and Chukotka; moreover, its distribution covered Western Siberia and the Ural Mountains [2,3].

The mitochondrial DNA haplotypes of modern brown bears are significantly correlated with the origin of the specimens, indicating probable maternal philopatry across the modern distribution of this species in the Holarctic [3,4,5,6].

Recent studies have shown that the mitochondrial genomes of brown bears that inhabited Northeastern Russia during the Pleistocene differ significantly from modern ones, which suggests a maternal component replacement at the turn of the Late Pleistocene—Early Holocene transition [3,7], due to significant ecosystem and climate changes that led to a mass Pleistocene megafauna extinction [9,10].

In the present study, we carry out a comparative paleogenetic analysis of the whole mitochondrial genome of a Holocene (3460 ± 40 years BP) brown bear fossil from Bolshoy Lyakhovsky Island (New Siberian Islands, Yakutia, Russia) and other ancient and modern brown bear specimens. We clearly demonstrate that the mitochondrial clade of modern Northeastern bears spread into this region no later than 3400 years ago. This result paves the way for subsequent work on fossilized permafrost brown bear samples. For example, this will enable us to assess the replacement of their mitochondrial lineages caused, apparently, by different food predispositions and other genetically determined phenotypical and physiological traits in *U. arctos* during the Late Pleistocene—Early Holocene transition.

## 2. Materials and Methods

### 2.1. Brown Bear Specimen Description and ^14^C Dating

The brown bear carcass, museum number: MM-F54 (Figure 1A and Appendix A), was found on Bolshoy Lyakhovsky Island (Yakutia, Russia) (Figure 1B) by a local herder and transferred to the Laboratory of P.A. Lazarev Mammoth Museum of the Research Institute of Applied Ecology of the North (Yakutsk, Russia). The brown bear carcass is partially mummified and subjected to cold sublimation. The mummy’s weight is 77.6 kg and its lifetime weight is estimated as >80 kg. Its length is about 155 cm, the height at the withers is about 75 cm, and the length of the hindfoot is 23.8 cm. The same size and weight are typical for young bears in the third year of life [11,12]. The preliminary morphological analysis of this brown bear also points to the young age of the animal. A winter fur coat covered the brown bear carcass at the moment of death, implying that the animal died during the cold season. Additional studies of the brown bear carcass are needed, including determination of sex and age.

The radius bone of the right limb of MM-F54 brown bear specimen was radiocarbon dated at the Carbon Analysis Lab Co., Ltd. (Gyeryong, Korea), based on the acid-based collagen-extracted fraction. The collagen was ^14^C dated by Accelerator Mass Spectrometry (AMS).

### 2.2. Ancient DNA Extraction, Library Preparation and DNA Sequencing

Ancient DNA was extracted from the brown bear skeletal muscle sample at the ancient DNA facilities of the National Research Center “Kurchatov Institute” (Moscow, Russia), following the previously described methodology [13,14]. This methodology includes several steps:Ancient DNA extraction using proteinase K in buffer containing EDTA.Ancient DNA enrichment on silica beads in binding buffer (contains tris(hydroxymethyl)aminomethane (TRIS), sodium acetate guanidine thiocyanate, and sodium chloride).Ancient DNA washing using ethanol.Ancient DNA elution in low-salt buffer.

Four independent ancient DNA extractions (in three replicates per skeletal muscle sample) were carried out. DNA quantity and quality were determined with Qubit 3.0 (Thermo Fisher Scientific, Waltham, WA, USA). Four multiplexed DNA libraries were prepared from the extracts with the highest DNA concentration using an Ovation® Ultralow Library System V2 kit (Tecan Genomics, Redwood, CA, USA). DNA libraries were constructed in ancient DNA facilities and final amplification of DNA libraries was performed in the modern DNA facility of the National Research Center “Kurchatov institute” (Moscow, Russia). The amplified DNA libraries were quantified using a high-sensitivity chip on a 2100 Bioanalyzer instrument (Agilent Technologies, Santa Clara, CA, USA). Multiple negative controls were used during the ancient DNA extraction and DNA-library amplification. The negative controls did not contain DNA after the DNA extraction, and the DNA libraries, which were prepared from the negative controls, were not amplified. Controls were not used for subsequent DNA sequencing.

Four multiplexed DNA libraries were test-sequenced on the S2 flowcell of the Illumina Novaseq6000 genome analyzer (Illumina, San Diego, CA, USA) at the sequencing facility of Kurchatov Center for Genome Research (Moscow, Russia) with paired-end reads of 150 bp in length. A number of endogenous reads were measured for each DNA library based on the bioinformatics procedures described below. Three DNA libraries (Medv4, Medv5, and Medv6) that produced >43% endogenous sequence reads (Appendix A) were used for the deep sequencing with Illumina NextSeq 500/550 High Out-put v2.5 flow cell of Illumina NextSeq 550 genome analyzer (Illumina, San Diego, CA, USA) at the sequencing facility of Nord University (Bodø, Norway) with single-end reads of 75 bp length.

### 2.3. Bioinformatics Analysis

Raw sequencing data trimming was performed with default parameters using the AdapterRemoval2 tool (version 2.2.2) [15]. Sequencing data were processed through the PALEOMIX 1.2.14 pipeline [16], including the merging of paired reads and mapping using BWA v0.7.17 with “rescale” option steps [17]. Mapping was performed against the brown bear reference genome sequence (assembly UrsArc1.0). Postmortem DNA damage patterns were analyzed using the MapDamage v2.0 tool [18]. We used MapDamage models to downscale the base quality scores according to the probability of DNA damage by-products to reduce the impact of nucleotide misincorporations in the downstream analyses.

Filtered genomic data were used for de novo assembly of the brown bear mitochondrial genome using SPAdes v3.15.3 [19]. The mitochondrial DNA sequence of *U. arctos* was extracted from the genome assembly using blastn 2.7.1+ and NCBI database. The resulting mitogenome sequence was annotated using MITOS [20]. The obtained annotation was then used to define partitions in the subsequent phylogenetic analysis.

Phylogenetic analysis of the mitochondrial DNA sequences was performed for the Pleistocene-Holocene Eurasian and North American brown bear specimens (Appendix A). Polar (*U. maritimus*) and cave bear (*U. spelaeus*) mitochondrial genomes were used as outgroups. The multiple sequence alignment was obtained with ClustalW using gap open penalty (15.00) and gap extension penalty (6) parameters [21]. The maximum likelihood (ML) analysis was conducted using RaxML v8.2.12. Maximum likelihood analysis was performed using the Tamura-Nei model; nodal support was evaluated using 100 replications of rapid bootstrapping implemented in RaxML [22]. The maximum parsimony analysis was also performed using the Subtree-Pruning-Regrafting model with 100 iterations of bootstrap testing. Phylogenetic tree reconstructions were drawn in iTOL v4 [23].

## 3. Results

### 3.1. AMS Radiocarbon Dating

The Holocene brown bear specimen, which was collected on Bolshoy Lyakhovsky Island (Yakutia, Russia) (Figure 1A), is stored in P.A. Lazarev Mammoth Museum (Yakutsk, Russia). The collagen tissue revealed a finite age: 3460 ± 40 years BP (IAEA C7-1 and IAEA C8-1) (Figure 1C).

### 3.2. Ancient DNA Sequencing and Brown Bear Mitogenome Assembly

We selected DNA libraries with relatively high endogenous DNA content from the test-sequencing run (paired-end, 2 × 150 bp); the total number of DNA reads generated for four *U. arctos* DNA libraries varied from 14,525,847 to 36,593,419 per library. The number of endogenous reads was measured for each DNA library with the PALEOMIX v1.2.14 pipeline [16]. The obtained DNA reads show an increased frequency of cytosine deamination substitutions at the 5′-ends of the sequenced DNA fragments, one of the traits of ancient DNA (Appendix A). The endogenous DNA content varied from 32.82 to 45.98 percent among the DNA libraries (Appendix A).

Three DNA libraries (Medv4, Medv5, and Medv6) with a high percentage of ancient DNA reads (>43%) were used for the deep sequencing (single-end sequencing, 75 bp). The total number of reads generated from these pooled brown bear DNA libraries was 425,509,359. These reads, as well as those previously obtained during the test-sequencing run, were used for de novo assembly of the mitogenome of the brown bear (Appendix A).

The mitogenome sequence of *U. arctos* was extracted from the assembled contigs using blastn 2.7.1+. The mitogenome of the brown bear consists of 16,654 bp (GenBank accession number: OP270839) and includes 22 tRNA, 2 ribosomal RNA and 13 protein-coding genes (Figure 2A). The order of genes and the direction of their transcription in the brown bear mitogenome are identical to the mitochondrial genomes of other, higher mammals [24].

### 3.3. Phylogenetic Analysis of Holocene Specimen and Other Ancient and Modern Brown Bears from Eurasia and North America Based on Their Mitochondrial Sequences

To evaluate the phylogenetic position of the Holocene brown bear specimen from the Bolshoy Lyakhovsky Island among other brown bears, which currently inhabit or previously inhabited Eurasia and North America (published data from previous studies), we generated a phylogenetic hypothesis based on whole mitochondrial DNA sequences (Appendix A). Phylogenetic trees based on maximum likelihood (Figure 2B) and maximum parsimony methods (Appendix A) have a similar topology.

Our phylogenetic reconstruction shows that the Holocene brown bear from Bolshoy Lyakhovsky Island (dated as 3460 ± 40 years BP) is clustered together with modern Northeast Eurasian brown bears and does not represent a separate mitochondrial lineage, together with previously published Middle and Late Pleistocene extinct *U. arctos* individuals from Yakutia and Chukotka [3,7].

## 4. Discussion

DNA extracted from museum samples is an important resource for understanding human- and environmental-caused changes in different animal species [9,25,26]. Mass extinction of species in the Late Pleistocene—Early Holocene appears to be due to drastic environmental and climate changes [10]. Nevertheless, the brown bear survived during this unstable climatic period, possibly due to its environmental plasticity, wide distribution, polyphagia, interspecific introgression with other bear species and other factors [27,28,29].

Modern brown bears mainly inhabit forest biotopes and they are very rarely observed in the tundra. Moreover, modern brown bears have never been observed on the New Siberian Islands. The bear sequenced in this study lived during one of the Holocene climatic optimums, the sub-boreal period when forest biotopes moved 150–300 km north. It remains a mystery how the studied bear reached the island, since between 4 and 7 thousand years ago, the land between Bolshoy Lyakhovsky Island and the mainland was inundated by the Arctic Ocean [30]. We assume that the brown bear arrived on the Bolshoy Lyakhovsky Island, which is located ~50 km from the mainland, during early spring, across the ice. At least, this is indicated by its winter fur coat and the recently recorded description of a brown bear on Wrangel Island (Chukotka, Russia), which is located approximately 145 km north of the mainland [31]. It seems that the North American and Eurasian Arctic can become a place of hybridization of polar and brown bears in a changing climate, as was described previously for Canadian bears [32]. Recent studies on the Pleistocene polar bear genomes also suggest several ancient introgression events with possible bidirectional gene flow between *U. arctos* and *U. maritimus* species [28,33]. Thus, introgressive hybridization between these two species occurred during the Pleistocene and continues nowadays. Moreover, polar bears carry a mitochondrial genome showing evidence of ancient hybridization with a brown bear [34], thus pointing to the paraphyletic type of evolution of brown bears (Figure 2B).

In this study, we present the complete sequence of the mitochondrial genome of a Holocene brown bear dated 3460 ± 40 years BP. Our comparative mitogenome analysis of Pleistocene and Holocene brown bears reveals the maternal philopatry of the Northeast Eurasian brown bear population during the last 3400 years. Moreover, the Northeast Eurasian brown bears (modern and Holocene one) are genetically close to the U16 Kol (4.2–3.6 kya) brown bear specimen from the foothills of the Altai Mountains (South Siberia) and have a common ancestor with modern *U. arctos* populations from Estonia and the European part of Russia (Figure 2B). Interestingly, the previously published Pleistocene brown bear specimen from South Siberia (U2 Chu, ~40 kya) [6] is genetically close to the Pleistocene specimens from Northeast Eurasia (F-2374, 25.9 kya and F-2296, 41.1 kya) [3] (Figure 2B).

Finally, we expect that the finding of additional brown and polar bear fossil specimens, the numbers of which are regularly increasing due to the effects of global warming on melting of permafrost, will allow for us to determine the approximate time of mitochondrial lineage replacement of the brown bear in Northeast Eurasia and its possible reasons. The Ursidae fossils from the Pleistocene and Holocene and their whole-genome analysis (including the brown bear specimen described in this study) will shed light on the possibility of introgressive hybridization between brown and polar bears in the Holarctic during these two geological epochs.

## Figures and Tables

**Figure 1 genes-13-01961-f001:**
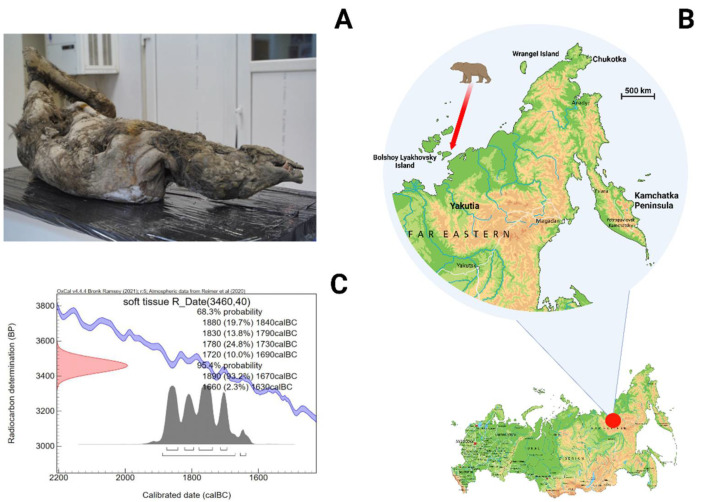
Brown bear specimen. (**A**) Frozen carcass of a brown bear fossil (MM-F54) in the Laboratory of P.A. Lazarev Mammoth Museum of the Research Institute of Applied Ecology of the North. (**B**) Geographical map showing the place where the brown bear carcass (MM-F54) was collected. (**C**) Calibrated ^14^C date for Holocene brown bear specimen, which was collected on Bolshoy Lyakhovsky Island (Yakutia, Russia).

**Figure 2 genes-13-01961-f002:**
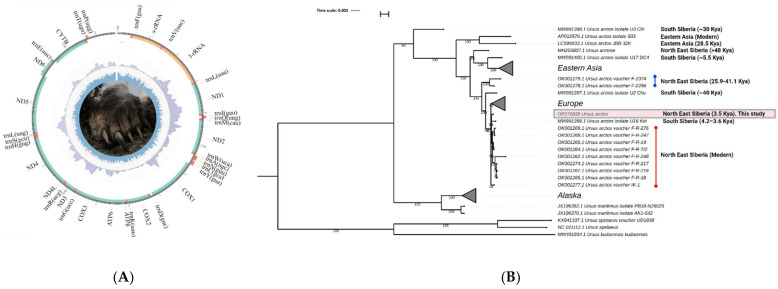
The mitochondrial genome of Holocene brown bear from Bolshoy Lyakhovsky Island (Yakutia, Russia). (**A**) Circular representation of the *U**. arctos* mitochondrial genome. Genes encoded by the heavy (H) strand are shown outside and those encoded by the light (L) strand are shown inside the mitogenome. Different genes are shown as filled boxes in different colours. The inner blue circle represents the GC content. The inner grey circle represents the mitogenome coverage. The brown bear limb (specimen number: MM-F54) is shown in the centre of the Figure. (**B**) Maximum likelihood phylogenetic tree of Ursus species, including the Holocene brown bear from Bolshoy Lyakhovsky Island (marked by a red box), based on their mitochondrial DNA sequences. Bootstrap values are shown in the nodes of the tree.

## Data Availability

The mitochondrial genome assembly of Holocene brown bear from Bolshoy Lyakhovsky Island is available for download through the National Center for Biotechnology Information (NCBI)—OP270839. Accession numbers of previously published *Ursus* specimens that were implemented in this study are available in Appendix A.

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
