# Peer review of "Ancient DNA Reveals Maternal Philopatry of the Northeast Eurasian Brown Bear (Ursus arctos) Population during the Holocene"

_genes, 2022, doi:10.3390/genes13111961_

Round 1

Reviewer 1 Report

The authors report a single mitochondrial genome from a 3,500-year-old brown bear mummy from Bolshoy Lyakhovsky Island. They obtained this data by shotgun sequencing and perform a single phylogenetic analysis with it. While the finding of the carcass is potentially interesting, since it is the first brown bear ever recovered from Bolshoy Lyakhovsky Island, the genetic results are really of limited interest, since female philopatry is nothing surprising for brown bears. However, given the endogenous content of >43% and the fact that a high-output sequencing kit was used that yielded, according to the authors 425 million reads, one can estimate that the authors should sit on data corresponding to at least a 3x nuclear genome. I explicitly do not want to speculate about the motivations why the authors did not include the results obtained from analyses of the nuclear data in this current manuscript, but simply advise them to rewrite the manuscript including the nuclear data, since these are much more informative regarding genetic continuity.

Minor points:

Abstract: “mass Pleistocene megafauna extinction” reads a bit strange. Maybe the authors could rephrase this.

Also, I am not sure if I agree with the statement that the brown bear survived “due to its environmental plasticity, polyphagia and genetic factors.” This statement is a bit too apodictic as we actually really do not know why some species survived and others went extinct. In the Discussion, introgressive hybridization is added as another explanation, although it is unclear how this relates to the results reported in the manuscript.

Generally, the English writing reads a bit clumsy. I do not know if any of the authors is a native speaker. If not, they might consider consulting a colleague who is and who can edit the writing.

M&M: I am not sure what the authors means with the small size of the upper canine from which they conclude that the animal was of young age. In mammals, teeth do not grow after they erupted fully. Thus, the only finding that can hint at a young age would be a canine that is not yet fully erupted. The size of a fully erupted tooth says nothing about the age of a bear.

Line 150: I think 3460 years is very much a finite, not an infinite age.

Author Response

We would like to thank you for your consideration, valuable comments, and suggestions to improve this manuscript. Corrections in manuscript are marked by yellow.

The authors report a single mitochondrial genome from a 3,500-year-old brown bear mummy from Bolshoy Lyakhovsky Island. They obtained this data by shotgun sequencing and perform a single phylogenetic analysis with it. While the finding of the carcass is potentially interesting, since it is the first brown bear ever recovered from Bolshoy Lyakhovsky Island, the genetic results are really of limited interest, since female philopatry is nothing surprising for brown bears. However, given the endogenous content of >43% and the fact that a high-output sequencing kit was used that yielded, according to the authors 425 million reads, one can estimate that the authors should sit on data corresponding to at least a 3x nuclear genome. I explicitly do not want to speculate about the motivations why the authors did not include the results obtained from analyses of the nuclear data in this current manuscript, but simply advise them to rewrite the manuscript including the nuclear data, since these are much more informative regarding genetic continuity.

Thank you for your suggestions. We are definitely going to analyze the nuclear genome of the Holocene brown bear in comparison with other brown bear genomes as well as ancient bear genomes (including Pleistocene polar bear and cave bear). This next step of our study will take more time, so we decided to start with the mitochondrial genome. We believe that the publication of this short paper per se is warranted, since modern and Pleistocene mitochondrial brown bear ancestry in Northeast Eurasia differs significantly.

Minor points:

Abstract: “mass Pleistocene megafauna extinction” reads a bit strange. Maybe the authors could rephrase this.

L31-L33: The sentence was rewritten as: “Significant palaeoecological and paleoclimatic changes which took place during Late Pleistocene – Early Holocene transition are considered important factors that led to megafauna extinctions.”

Also, I am not sure if I agree with the statement that the brown bear survived “due to its environmental plasticity, polyphagia and genetic factors.” This statement is a bit too apodictic as we actually really do not know why some species survived and others went extinct. In the Discussion, introgressive hybridization is added as another explanation, although it is unclear how this relates to the results reported in the manuscript.

L33-L34: The sentence was rewritten as: “Unlike many other species, the brown bear (Ursus arctos) has survived this geological time”. The sentence about introgressive hybridization was excluded.

L224-L235: A significant part of the Discussion was rewritten. Hybridization between polar and brown bears is also discussed because it is important for understanding the adaptive potential of mammalian species to climate change.

Generally, the English writing reads a bit clumsy. I do not know if any of the authors is a native speaker. If not, they might consider consulting a colleague who is and who can edit the writing.

The manuscript has been carefully proofread by a native English speaker.

M&M: I am not sure what the authors means with the small size of the upper canine from which they conclude that the animal was of young age. In mammals, teeth do not grow after they erupted fully. Thus, the only finding that can hint at a young age would be a canine that is not yet fully erupted. The size of a fully erupted tooth says nothing about the age of a bear.

L85-L89: Thank you for the correction. We rewrote this section accordingly.

Line 150: I think 3460 years is very much a finite, not an infinite age.

L163: The correction was added.

Reviewer 2 Report

The authors sequenced the complete mitogenome of the brown bear from a mummified carcass of about 3400-year-old and tried to explain its phylogeny with recent as well as earlier sequences of ancient samples. This paper is nicely represented the evolution pattern of modern U. arctos.     

In figure 2, U. arctos clade is paraphyletic that needs further explanation in the discussion and conclusion sections.

Line 150: Place ‘.’ after Russia)

Line 143: In place of maximum likelihood (ML) analysis, authors need to use a patrimony tree for such important interferences.

Line 222: ‘Last but not the least, a recent study…..’ seems like an acknowledgment paragraph, which needs to be re-written as per standard conclusion format. The conclusion portion also needs to be elaborated for providing more information.

Author Response

We would like to thank you for your consideration, valuable comments, and suggestions to improve this manuscript. Corrections in manuscript are marked by yellow. Answers are in blue.

The authors sequenced the complete mitogenome of the brown bear from a mummified carcass of about 3400-year-old and tried to explain its phylogeny with recent as well as earlier sequences of ancient samples. This paper is nicely represented the evolution pattern of modern U. arctos.    

In figure 2, U. arctos clade is paraphyletic that needs further explanation in the discussion and conclusion sections.

L233-L235: Thank you for your suggestion. The evolution of this paraphyletic clade is now discussed

Line 150: Place ‘.’ after Russia)

Corrected.

Line 143: In place of maximum likelihood (ML) analysis, authors need to use a patrimony tree for such important interferences.

The specimens of the modern North Eastern brown bear population as well as ancient brown bears from the Siberia and adjacent regions are marked on the phylogenetic tree. A parsimony phylogenetic tree was additionally constructed and included as Supplementary Material (Figure S3) based on your recommendation.

Line 222: ‘Last but not the least, a recent study…..’ seems like an acknowledgment paragraph, which needs to be re-written as per standard conclusion format. The conclusion portion also needs to be elaborated for providing more information.

L224-L254: A significant part of the Discussion was rewritten based on yours and the previous reviewer’s comments.

Reviewer 3 Report

Thank you to the authors for this interesting contribution! I enjoyed the article overall. The methods were appropriate, and the implications and scope of the paper are well stated, but I think that improving the writing will make it easier to read. I made several line-specific comments below, but in general, the authors could please add more details to their methods section, remove methods from the results section, and further develop/add to the introduction and discussion without repeating lines in these two sections.

Line edits:

Line 36. Please edit to “…is still widely distributed in Northeast Eurasia.”

Line 47. For those who may be unfamiliar, please indicate what the Yakutia, Chukotka, and Kamchatka are.

Line 55. Please consider editing to “Brown bears have inhabited Northeast Eurasia since at least the Middle Pleistocene…” or similar.

Line 60. Please replace “assuming” with “indicating probable maternal philopatry” or similar.

Line 65. Please edit to …”due to significant ecosystem and climate changes that led to a mass Pleistocene megafauna extinction.”

Line 70. Please do not capitalize Specimens here.

Line 106. Please be more precise here – how may extractions did you perform in total? How did you quantify the DNA prior to library prep? Was the library prep conducted in the ancient DNA lab? Did you sequence the negative control? There must have been some DNA there, at least dimer?

Line 113. Which sequencing core did you use? Or was the NovaSeq at your institution?

Line 115. I think you don’t need this italicized line, the reader will get to that later, and you can save words by removing it.

Line 117. Please clarify that you deeply sequenced the three libraries that produced >43% endogenous sequence reads.

Line 127. Please edit to “…, including the merging of paired reads and mapping using…”

Line 127. Please indicate what you used to check for/remove any contaminating adapter sequences.

Line 130. Please edit this line for clarity or remove it.

Line 142. Please edit to, “…were used as outgroups.” Also please indicate what program/algorithm that you used for multi-sequence alignment.

Line 144. Please add information about what model of molecular evolution you used. Did you perform model testing?

Line 148. I don’t think that you need to restate the methods here, and you can simply state the results of the carbon dating analysis.

Line 153. Again, I think you don’t need to summarize the method again, just indicate how many libraries you successfully sequenced, the percent endogenous content (what was the range of values?), and then continue with the next paragraph.

Line 162. Please edit to replace “polled brown bear libraries” with “pooled brown bear libraries”

Lines 165-169. In these paragraphs as well, please remove methods.

Line 178. Instead of using the / in inhabite/-ed please consider rewriting this sentence to something like, “…among other brown bears that currently inhabit or previously inhabited Eurasia and North America.”

Line 180. Please edit to, “…,we generated a phylogenetic hypothesis…” or similar – phylogenetic trees are based on models of evolution and aren’t technically clustering analyses (like distance-based methods).

Line 209. Please edit this sentence for clarity. Perhaps consider something like, “The bear sequenced in this study lived during one of the Holocene…”

Line 210. How far is the island from the mainland? Perhaps you can mention/compare the distance between the Admiralty Islands and mainland Alaska (presumable brown bears were able to swim to colonize these islands). How far are brown bears known to swim?  

Line 220. Could you add more discussion about the replacement of these lineages? Where did the bears come from that replaced the bears that occupied this region previously? Is your sequence closer to a specific European lineage? Maybe speculate a little here.

Line 222. Consider adding a second citation to the other Pleistocene bear genome paper that came out at the same time (Wang, MS., Murray, G.G.R., Mann, D. et al. A polar bear paleogenome reveals extensive ancient gene flow from polar bears into brown bears. Nat Ecol Evol 6, 936–944 (2022). https://doi.org/10.1038/s41559-022-01753-8). Include more discussion about exactly how nuclear DNA from the bear you sequenced here can contribute to our understanding of bear evolution. 

Figures.

Figure 1.A. Wow that picture is impressive!! Do you have any other angles that you could provide? Perhaps in the supplement? 1.B. For those who aren’t familiar with the region, perhaps consider including an inset map showing the location of the island relative to Russia/Alaska? A scale bar is also needed in the map.

Figure 2.A. For the final publication, please make sure this figure is sufficient quality so that the gene names can be read. Figure 2.B. It looks like in your tree, one of the polar bear sequences does not cluster with the others – is this a mistake? Should it be U. arctos? Also please add more information to make this tree easier to interpret. Which sequences belong to the lineage that was displaced that used to inhabit NE Russia? Are they the lineages at the top of the tree? Maybe one way to do this is to indicate the sampling locality of each bear at the tip of the tree itself, or to edit the figure to put bars or shading to show this information?

Author Response

We would like to thank you for your consideration, valuable comments, and suggestions to improve this manuscript. Corrections in manuscript are marked by yellow. Answers are in blue.

Thank you to the authors for this interesting contribution! I enjoyed the article overall. The methods were appropriate, and the implications and scope of the paper are well stated, but I think that improving the writing will make it easier to read. I made several line-specific comments below, but in general, the authors could please add more details to their methods section, remove methods from the results section, and further develop/add to the introduction and discussion without repeating lines in these two sections.

First of all, thank you for the detailed review. Your valuable comments and suggestions helped us to improve manuscript significantly.

Line edits:

Line 36. Please edit to “…is still widely distributed in Northeast Eurasia.”

L35: Corrected.

Line 47. For those who may be unfamiliar, please indicate what the Yakutia, Chukotka, and Kamchatka are.

L45-L46: Corrected. These geographical regions were also added in Figure 1B. Yakutia, Chukotka, Kamchatka Peninsula and even Wrangel Island were marked on the map.

Line 55. Please consider editing to “Brown bears have inhabited Northeast Eurasia since at least the Middle Pleistocene…” or similar.

L54: Corrected.

Line 60. Please replace “assuming” with “indicating probable maternal philopatry” or similar.

L59: Corrected.

Line 65. Please edit to …”due to significant ecosystem and climate changes that led to a mass Pleistocene megafauna extinction.”

L64: Corrected.

Line 70. Please do not capitalize Specimens here.

L69: Corrected.

Line 106. Please be more precise here – how may extractions did you perform in total? How did you quantify the DNA prior to library prep? Was the library prep conducted in the ancient DNA lab? Did you sequence the negative control? There must have been some DNA there, at least dimer?

L106-L109: The section about DNA extraction and DNA library preparations was improved. We did not sequence negative controls, since they had not contained a significant amount of DNA molecules after cleaning of PCR mixes.

Line 113. Which sequencing core did you use? Or was the NovaSeq at your institution?

L119-L120 and L125-L126: Information about sequencing facilities was added.

Line 115. I think you don’t need this italicized line, the reader will get to that later, and you can save words by removing it.

L122: Amended.

Line 117. Please clarify that you deeply sequenced the three libraries that produced >43% endogenous sequence reads.

L122-L123: We added an explanation that Medv4, Medv5, and Medv6 DNA libraries were used for the deep sequencing.

Line 127. Please edit to “…, including the merging of paired reads and mapping using…”

L137-L138: Corrected.

Line 127. Please indicate what you used to check for/remove any contaminating adapter sequences.

L135-L136: The raw sequencing data trimming was performed with default parameters using the AdapterRemoval2 tool.

Line 130. Please edit this line for clarity or remove it.

We decided to exclude this sentence.

Line 142. Please edit to, “…were used as outgroups.” Also please indicate what program/algorithm that you used for multi-sequence alignment.

L151-L152: Corrected.

L152-L153: Information about multi-sequence alignment was included in the Methods section.

Line 144. Please add information about what model of molecular evolution you used. Did you perform model testing?

L154-L158: Information about molecular evolution models was included to the Methods section.

Line 148. I don’t think that you need to restate the methods here, and you can simply state the results of the carbon dating analysis.

Amended.

Line 153. Again, I think you don’t need to summarize the method again, just indicate how many libraries you successfully sequenced, the percent endogenous content (what was the range of values?), and then continue with the next paragraph.

The beginning of the Results section was modified as suggested.

Line 162. Please edit to replace “polled brown bear libraries” with “pooled brown bear libraries”

Corrected. 

Lines 165-169. In these paragraphs as well, please remove methods.

This sentence was partially excluded. The part which describes cytosine deamination substitutions at the 5′-ends in DNA libraries is moved to L170-L171.

Line 178. Instead of using the / in inhabite/-ed please consider rewriting this sentence to something like, “…among other brown bears that currently inhabit or previously inhabited Eurasia and North America.”

L189-L190: Corrected. 

Line 180. Please edit to, “…,we generated a phylogenetic hypothesis…” or similar – phylogenetic trees are based on models of evolution and aren’t technically clustering analyses (like distance-based methods).

L190: Amended. 

Line 209. Please edit this sentence for clarity. Perhaps consider something like, “The bear sequenced in this study lived during one of the Holocene…”

L220: Amended. 

Line 210. How far is the island from the mainland? Perhaps you can mention/compare the distance between the Admiralty Islands and mainland Alaska (presumable brown bears were able to swim to colonize these islands). How far are brown bears known to swim?

L224-L226: We suppose that bear could have walked across the ice. We added the distance from Bolshoy Lyakhovsky Island to the mainland (~50 km).

L226-L228: We also added information that the brown bear came to Wrangel Island in 2019 from the mainland (145 km).

Line 220. Could you add more discussion about the replacement of these lineages? Where did the bears come from that replaced the bears that occupied this region previously? Is your sequence closer to a specific European lineage? Maybe speculate a little here.

L237-L247: A significant part of the Discussion was rewritten, including a comparison with the European mtDNA lineage.

Line 222. Consider adding a second citation to the other Pleistocene bear genome paper that came out at the same time (Wang, MS., Murray, G.G.R., Mann, D. et al. A polar bear paleogenome reveals extensive ancient gene flow from polar bears into brown bears. Nat Ecol Evol 6, 936–944 (2022). https://doi.org/10.1038/s41559-022-01753-8). Include more discussion about exactly how nuclear DNA from the bear you sequenced here can contribute to our understanding of bear evolution. 

L232: This citation was added.

L248-L255: We expanded the part which explains why we need more mitochondrial and nuclear genomes from ancient brown and polar bears in the Holarctic.

Figures.

Figure 1.A. Wow that picture is impressive!! Do you have any other angles that you could provide? Perhaps in the supplement? 1.B. For those who aren’t familiar with the region, perhaps consider including an inset map showing the location of the island relative to Russia/Alaska? A scale bar is also needed in the map.

Figure 1 was updated based on your suggestions. Figure S1 contains additional angles on the brown bear which was found in 2020: https://www.bbc.com/news/world-europe-54160645. Yakutia, Chukotka, Kamchatka Peninsula and even Wrangel Island were marked on the map. A scale was also added.

Figure 2.A. For the final publication, please make sure this figure is sufficient quality so that the gene names can be read. Figure 2.B. It looks like in your tree, one of the polar bear sequences does not cluster with the others – is this a mistake? Should it be U. arctos? Also please add more information to make this tree easier to interpret. Which sequences belong to the lineage that was displaced that used to inhabit NE Russia? Are they the lineages at the top of the tree? Maybe one way to do this is to indicate the sampling locality of each bear at the tip of the tree itself, or to edit the figure to put bars or shading to show this information?

Figure 2A was improved: The font size for gene names and image quality were changed.

Figure 2B was improved: The polar bear among brown bears was related to a typo in NCBI from our colleagues. In the paper by Kosintsev et al (2022) the bear (F-2374) was marked as U. arctos, but in NCBI it was marked as U. maritimus: https://www.ncbi.nlm.nih.gov/nuccore/OK001279.1/. We contacted the authors and pointed out the typo.

The specimens of the modern North Eastern brown bear population as well as ancient brown bears from Siberia and adjacent regions are marked on the phylogenetic tree. A parsimony phylogenetic tree was additionally constructed and included in the Supplementary (Figure S3) based on the recommendation of Reviewer #2.